# Short Linear Motifs Characterizing Snake Venom and Mammalian Phospholipases A2

**DOI:** 10.3390/toxins13040290

**Published:** 2021-04-20

**Authors:** Caterina Peggion, Fiorella Tonello

**Affiliations:** 1Department of Biomedical Sciences, University of Padova, Via U. Bassi, 58/B, 35131 Padova, Italy; caterina.peggion@unipd.it; 2CNR of Italy, Neuroscience Institute, viale G. Colombo 3, 35131 Padova, Italy

**Keywords:** snake venom phospholipases A2, neurotoxins, myotoxins, secretory phospholipases A2, PLA2G1B, PLA2G2A, sequence alignment, short linear motifs, prolyl isomerase, glycogen synthase kinase 3

## Abstract

Snake venom phospholipases A2 (PLA2s) have sequences and structures very similar to those of mammalian group I and II secretory PLA2s, but they possess many toxic properties, ranging from the inhibition of coagulation to the blockage of nerve transmission, and the induction of muscle necrosis. The biological properties of these proteins are not only due to their enzymatic activity, but also to protein–protein interactions which are still unidentified. Here, we compare sequence alignments of snake venom and mammalian PLA2s, grouped according to their structure and biological activity, looking for differences that can justify their different behavior. This bioinformatics analysis has evidenced three distinct regions, two central and one C-terminal, having amino acid compositions that distinguish the different categories of PLA2s. In these regions, we identified short linear motifs (SLiMs), peptide modules involved in protein–protein interactions, conserved in mammalian and not in snake venom PLA2s, or vice versa. The different content in the SLiMs of snake venom with respect to mammalian PLA2s may result in the formation of protein membrane complexes having a toxic activity, or in the formation of complexes whose activity cannot be blocked due to the lack of switches in the toxic PLA2s, as the motif recognized by the prolyl isomerase Pin1.

## 1. Introduction

Phospholipases (PLs) are fundamental lipolytic enzymes for living organisms. They are classified in different families (A1, A2, B, C and D), distinguishable by the position where they induce lipid hydrolysis [1,2]. The PLA2 family is involved in the cleavage of an sn-2 ester bond of the glycerophospholipid substrate, with the consequent generation of 1-acyl-lysophospholipids and free fatty acids, in particular polyunsaturated acids that are metabolized to eicosanoids and bioactive lipid mediators [3,4]. Their key role in inflammation processes and in the arachidonic acid metabolism in mammals is what made such enzymes the most studied PLs. Inside the PLA2 family, it is possible to discriminate different subfamilies based on their structure, localization, and catalytic mechanism [5]. One of these subfamily comprises the secretory PLA2s (sPLA2s) (EC 3.1.1.4) which is subdivided in seventeen groups [6], according to their molecular structures. Interestingly, sPLA2s are well-conserved from eubacteria to mammals, suggesting an early appearance during evolution [4,6,7]. Conventional sPLA2s (groups I, II, V and X) are extracellular small-molecular-weight (13–15 kDa) enzymes, requiring millimolar Ca^2+^ concentrations for their catalytic activity, principally targeting phospholipids in the extracellular milieu. sPLA2s are widely distributed in different body fluids (e.g., snake and other venoms, plasma, pancreatic juice, tears, seminal fluid) and are involved in a variety of processes, from snakebite envenomation to signal transduction, to lipid mediator production and dietary lipids digestion in mammals [6,8,9]. Moreover, it is important to consider that snake venom PLA2s toxins are potential therapeutic drugs against many pathophysiological conditions [10].

More than 400 unique snake venom PLA2s toxins are reported in the UniProtKB database, with an overall conserved tertiary structure [11]. All these sPLA2s toxins represent optimal models for the study of sPLA2 function and mechanism of action. Based on primary structure similarity, position of disulfide bridges, and loop insertion, snake venom PLA2s are separated in two groups. Those derived from Elapidae and Hydrophiidae snakes belong to the group I PLA2s, more similar to the mammalian pancreatic phospholipases (PLA2G1B), while those deriving from Viperidae and Crotalidae snakes belong to group II, given their high homology with the mammalian phospholipases enriched in synovial fluids and tears (PLA2G2A) [10,12]. Snake venom PLA2s are also subdivided in three different categories based on their toxic effects: myotoxins, targeting skeletal muscles; neurotoxins, that act on pre- or post-synaptic elements of the neuromuscular junction; and hemostasis-impairing toxins [12,13]. 

sPLA2s of both groups share a highly conserved amino acidic sequence and a well-conserved tertiary structure (Figure 1), that was unveiled by the crystallization of various sPLA2s isolated from snake venoms, humans, and bovines [4,6,7]. Although the majority of snake venom PLA2s exist as monomers, some of them have acquired a quaternary structure by a non-covalent interaction between two or more PLA2s [12].

sPLA2s are characterized by the presence of three α-helixes, a highly conserved calcium binding loop, two anti-parallel β-strands, and a flexible C-terminal loop [14]. sPLA2s are also characterized by a well-conserved His-Asp active site in the catalytic motif, in which the Asp is implicated in the coordination of Ca^2+^ with the Tyr and Gly amino acids in the Ca^2+^ binding motif, and extensively conserves disulfide bonds that ensure the stabilization of the tertiary structure [12,15]. Group I PLA2s is distinguishable from group II PLA2s for the presence of the so-called elapid loop located between the second α-helix and the β-strands, and of a different disulfide bond pattern [6]. Unfortunately, a correlation between the toxic effects of different snake venom PLA2s and their primary structure is still unknown. Moreover, many group II snake venom PLA2s are catalytically inactive due to a missense mutation of D49 amino acid (more frequently substituted with a lysine) [12,16], but despite this mutation, these PLA2-like proteins are highly toxic, mostly myotoxic, even more than their catalytically active counterparts [17]. Therefore, the emerging idea is that the sPLA2 toxic activities depend not only on their enzymatic functions, but also by their interaction with target proteins that may mediate their entry into the cells, activating different signaling cascades, and their intracellular activity. 

In this complicated context, understanding the molecular mechanisms of toxin actions may help not only to clarify steps of snakebite envenomation, but also to shed light on the mode of action of mammalian sPLA2s homologues. Indeed, besides the biological functions of mammalian sPLA2s, it is now well recognized that their increased expression is related to pathological processes such as certain types of cancers, arthritis, and inflammatory disorders. For this reason, sPLA2s represent awesome therapeutic targets for the pharmaceutical industry [3,18,19,20,21]. 

As previously anticipated in the conference proceedings of “The 1st International Electronic Conference on Toxins (IECT 2021)”, here follows an implementation of the already reported in silico analysis of the structures of monomeric or homomeric snake myotoxic and neurotoxic PLA2s. This analysis was performed by comparing snake venom PLA2s with their mammalian counterparts, aimed to unveil exposed sites that may represent potential candidates for modification or interaction with other proteins. This analysis was made possible thanks to the “animal toxin annotation project” of the Swiss-Prot database, in which venom protein sequences are systematically curated to the standards of UniProtKB/Swiss-Prot [22]. From the analysis of the two high variable regions, comprising the second and third α-helixes and the C-terminal loop, we identified amino acidic enrichments that discriminated different families. Moreover, by analyzing the sequences for the presence of short linear motifs (SLiMs), via the Eukaryotic Linear Motif (ELM) resource [23,24], we identified well-conserved phosphorylable sites, ligand binding stretches, and other motifs characterizing the different subgroups of sPLA2s. 

## 2. Results

### 2.1. Sequence Alignment Comparison

We have collected, from UniProt, the curated entries of 4 myotoxins, 7 neuro-myotoxins, and 14 neurotoxins of group I snake venom PLA2s, 39 myotoxins (of which 14 of the group D49), 14 neuro-myotoxins (of which 9 of the group D49), and 12 neurotoxins of group II snake venom PLA2s. We have selected only monomeric or homomeric toxins. We decided to discard myotoxins of group I because there were not enough compared to the others. We then collected the ten manually reviewed entries of mammalian PLA2G1B, and the five manually reviewed entries of mammalian PLA2G2A plus five other computationally analyzed entries, to have the same number of sequences in the two mammalian groups.

We aligned the collected sequences with Clustal Omega (Appendix A), represented the alignments using the software Snapgene, and compared the alignments in Figure 2A,B. These proteins have very conserved sequence tracts: the calcium binding loop region (in grey in Figure 1 and Figure 2) and the second and third α-helixes that contain the amino acids involved in the active site (HD 48–49 and the aspartic acid of the motif C.CD). The first α-helix has a higher variability with respect to the other two α-helixes. The regions of greatest variability are the section between the first α-helix and the calcium binding loop, the central area between the second and third α-helixes, which also includes the β-sheet, and the C-terminal area after the third α-helix.

The calcium binding loop region contains the motif YGC[YNHFW]CG, present only in sPLA2s and in otoconins, proteins of otoconia, agglomerates of calcium salts in the inner ear [25]. Currently, 460 proteins reported in Swiss-Prot (the manually annotated section of UniprotKB) possess this motif: 458 sPLA2s and 2 otoconins. The variable amino acid of this pattern is almost always a Y in group I PLA2s, an H in mammalian group II PLA2s, and an N in myotoxins not-D49. In the latter case, there is no coordination of the calcium ion, due to the lack of the aspartic residue in position 49; therefore, the preservation of the asparagine must have another meaning. In some categories of sPLA2s, other amino acids are also conserved, before and after the motif YGC.CG. Glycine is also highly conserved, and is involved in the coordination of the calcium ion, in second position after the motif. 

In the active site pocket of the not-D49 toxins, the aspartic acid is usually substituted by a K (24/30); more rarely by R (2/30), S (2/30), N (1/30) or Q (1/30). The aspartic acid of the CxCD motif, present in the III α-helix and involved in the enzymatic catalysis, was conserved in all the phospholipases analyzed. It is curious that the x of this motif, in group I PLA2s, is always a D in the case of neurotoxins, and an N in mammalian sPLA2s, whereas in neuro-myotoxins the variability is higher. In group II, all toxins have an E in this position, while in mammalian sPLA2s both an E and a Q can be found.

Other residues conserved among sPLA2s of a single group are the N-terminal amino acid and the tryptophan. The N-terminal amino acid is always N in the group I toxic PLA2s, A in PLA2G1B, and S in toxins of group II not-D49. PLA2G1B has a conserved W in third position, while in the toxins of group I W is often found in position 18–19, and sometimes in 110. In mammalian PLA2G2A, W is mostly absent, while in snake venom group II, it is conserved in position 68 of not-D49 toxins and in position 30 of D49 toxins. Tryptophan is the least frequently occurring amino acid in proteins, and because of its properties, it can strongly influence interactions with the plasma membrane (PM) and other proteins. 

Concerning the two regions with the greatest variability, which are also the lower complexity regions (composed of fewer types of amino acids), we have performed an analysis of the amino acid composition which we report in the following paragraph. In the last paragraph of the results, we describe the SLiMs that we found to be uniquely present in some groups of sPLA2s.

### 2.2. Amino Acid Composition Analysis of the Central and C-Terminal Regions

As previously introduced, sPLA2s protein sequences are characterized by the presence of two low complexity regions, one included by the second and the third α-helixes and the other represented by the C-terminal stretch. These two regions are characterized by the enrichment of some types of amino acids and by a lower conservation across sPLA2s protein families (Figure 2A,B). To unveil motifs or amino acids more abundant in these two regions, that may be important for the biological roles of sPLA2s, we have evaluated the mean abundance of different categories of amino acids, taking into consideration particular amino acidic properties.

As shown in Figure 3A, the central region of sPLA2s (group I) myotoxins and neurotoxins (red and yellow bars) is characterized by a lower mean abundance of phosphorylable Ser/Thr amino acids and of negatively charged amino acids (D/E), with respect to mammalian PLA2G1B (green bars). 

Concurrently, an increased presence of all the other amino acids is observed in toxins of the group I PLA2s. Of these, glycine is the most represented (see Appendix A, Appendix A). Regarding the C-terminal region, it is possible to observe an increased amount of N/Q and a lower amount of charged amino acid (D/E and K/R) in toxins with respect to mammalian PLA2G1 (Figure 3B). The central region of group II PLA2s shows a pattern less distinctive for each family, with a mild increment of D/E and of phosphorylable Y in toxins with respect to mammalian PLA2G2A (green bars, Figure 3C). Conversely, more differences can be observed in the C-terminal region of sPLA2s (group II). In particular, the total absence of D/E in mammalian PLA2G2A is evident, as well as the lower amount of N/Q in toxins and of S/T in myotoxins and neuro-myotoxins (not-D49) (Figure 3D).

### 2.3. SLiMs Conserved in Toxins and Not in Mammalian PLA2s or Vice Versa

We searched the SLiMs present in the ELM resource [24] and conserved in the different groups of sPLA2s. We selected for the search “all cellular compartments” because sPLA2s are internalized in cells, and were observed to localize in different cellular organelles, in vesicles in the cytoplasm, in mitochondria, and in the nucleus [26,27,28,29,30]. Of all the motifs identified (Appendix A), we selected those conserved in toxins but not in mammalian PLA2s, or vice versa. Furthermore, to decrease the possibility of false positives, we only considered motifs present in exposed traits of the protein structure, obtaining the SLiMs described in Table 1, Table 2 and Table 3 and in the next paragraphs. 

In the description of proteins and cellular processes involving the described SLiMs, in Table 3 and in the following paragraphs, we have focused on events in which sPLA2s are typically involved, as membrane processes, lipid metabolism, and inflammation. 

The N-terminal asparagine of group I toxins is part of the N-degrons that cause the protein to undergo ubiquitin-dependent proteasomal degradation [31]. There is no apparent reason for toxins to benefit from the degradation process; perhaps this asparagine also has another function. Between the first alpha-helix and the calcium binding loop in PLA2G1B, two prolines are conserved, spaced by three amino acids and with a serine in the middle. This motif has never been described and associated to particular interactions or activities, but we think it is interesting for the presence of two prolines before and after a phosphorylable amino acid; moreover, this motif is not present in group I toxins where in its place an arginine residue is conserved.

The calcium binding loop region of snake venom and mammalian group I PLA2s differs by a leucine at position 31. This leucine is part of the motif recognized by the NimA-related kinases (NEKs) that phosphorylate S/T at position +3 relative to leucine (S/T 34 in mammalian group I PLA2s). NEKs are a large group of kinases known mainly for their role in the cell cycle, but are also involved in other processes including inflammation [32]. The pancreatic loop, characteristic of the mammalian PLAG1B, together with the following tyrosine (Y69), form the triplet NPY, a motif present in a pathogenic *Escherichia coli* effector protein, that binds to the I-BAR domain of a host protein and induces actin polymerization and the formation of the pedestal necessary for the infection process [24,33]. The Y69, when phosphorylated, is also a ligand for the Src homology 2 (SH2) domain of Signal Adaptor Protein 1 (STAP1), an adaptor protein that besides SH2 possesses a lipid-binding Pleckstrin homology (PH) domain [34]. 

In the beta-sheet region of PLA2G1B, a string of 7–9 serine or threonine amino acids alternating with other amino acids, is present (Figure 4A). This S/T string in toxins of the group I is not conserved, and in accordance the S/T content of the central region of these proteins is lower than in PLA2G1B (Figure 3); in some cases, such as notexin, only one S is present (Figure 4A). This region of PLA2G1B contains up to four motifs recognized by glycogen synthase kinase 3 (GSK3), a specific kinase for serine and threonine. The C-terminal part of this S/T series included, in five of ten of the mammalian PLA2G1B, a phosphorylation site for casein kinase 1 or 2 (CK1/2). GSK3 is a kinase that requires priming phosphorylation on one or more sites in the C-terminal with respect to its point of action, and CK1/2 or other kinases that phosphorylate the S/T in the C-terminal of the S/T string can initiate a series of phosphorylation by GSK3 [35]. GSK3 controls many biochemical pathways, mostly by inactivating its target proteins, and is involved in the regulation of inflammatory processes [36].
toxins-13-00290-t001_Table 1Table 1Short Linear Motifs (SLiM) conserved in exposed regions of snake venom Group I.MotifDescriptionELM ClassSequence SectionStart–EndFraction of Proteins Containing the Motif in the Specified SectionNeuro-MyotoxinsNeurotoxinsPLA2G1BN-terminalDEG_Nend_UBRbox_317/714/140/10NEK2 phosph. siteMOD_NEK2_131–360/70/149/10I-BAR binding siteLIG_IBAR_NPY_167–690/70/1410/10SH2 binding siteLIG_SH2_STAP169–730/70/1410/10GSK3 phosph. sites *MOD_GSK3_167–800/71/1410/10SH2 binding site LIG_SH2_NCK_1105–1093/76/140/10PTB binding sitesLIG_PTB_Apo_2LIG_PTB_Phospho_1104–1111/79/140/10* Three to four consecutives.
toxins-13-00290-t002_Table 2Table 2Short Linear Motifs (SLiM) conserved in exposed regions of snake venom Group II.MotifDescriptionELM ClassSequence SectionStart–EndFraction of Proteins Containing the Motif in the Specified SectionMyotoxinsNeuro-MyotoxinsNeurotoxinsPLA2G2ANOT D49D49NOT D49D49D49
Pin1 site, andPDK and CDK phosph. sitesDOC_WW_Pin1_4MOD_ProDKin_1MOD_CDK_SPK_235–370/250/141/51/94/1210/10SH2 binding siteLIG_SH2_CRK51–5523/259/143/56/99/120/10PKA phosph. siteMOD_PKA_152–5623/253/143/51/90/120/10FHA binding siteLIG_FHA_159–650/250/140/52/94/1210/10Pin1 site andCDK phosph. siteDOC_WW_Pin1_4MOD_CDK_SPK_2119–1210/250/140/50/90/127/10PDZ binding siteLIG_PDZ_Class_3119–12123/259/145/55/910/120/10
toxins-13-00290-t003_Table 3Table 3Short Linear Motifs (SLiM) description.ELM Class IdentifierRegular ExpressionInteraction Partner(s)Examples of Proteins * Containing the SLiMDEG_Nend_ UBRbox_3^M{0,1}([NQ])Aminohydrolases for deamidation-DOC_WW_Pin1_4...([ST])P.Peptidyl prolyl isomerase 1 ManyLIG_FHA_1..(T)..[ILV].Proteins containing the forkhead-associated domain, e.g., TIFA, TIFAB, AGGF1Kinesin interactors, TIFA LIG_IBAR_NPY_1NPYI-BAR domain-containing proteins, involved in membrane dynamic The bacterial protein Tir, SHANK2LIG_PDZ_Class_3...[DE].[ACVILF]$PDZ containing proteins, PDZ domains also bind to phospholipid headgroupsMany PDZ ligands are membrane proteins LIG_PTB_Apo_2LIG_PTB_Phospho_1(.[^P].NP.[FY].)|(.[ILVMFY].N..[FY].)(.[^P].NP.(Y))|(.[ILVMFY].N..(Y))Proteins containing phosphotyrosine binding (PTB) domains, e.g., insulin receptor substrate 1 (IRS-1)Integrins, LRP1LIG_SH2_CRK(Y)[^EPILVFYW][^HDEW][PLIV][^DEW]Proteins containing SH2 domain of the CRK familyTransmembrane receptors,Phospholipase C-gamma-1LIG_SH2_NCK_1(Y)[DESTNA][^GWFY][VPAI][DENQSTAGYFP]Proteins containing SH2 domain of the NCK familyTransmembrane proteins, adapter protein docking 1LIG_SH2_STAP1(Y)[DESTA][^GP][^GP][ILVFMWYA]Proteins containing SH2 domain of the STAP1 familyTransmembrane proteins,lipid phosphataseMOD_CDK_SPK_2...([ST])P[RK]Cyclin-dependent kinases, proline directed kinaseProteins involved in different biochemical pathwaysMOD_GSK3_1...([ST])...[ST]Glycogen synthase kinase 3, needs priming, inhibitoryMOD_NEK2_1[FLM][^P][^P]([ST])[^DEP][^DE]Never in mitosis A (NimA)-related kinasesMOD_PKA_1[RK][RK].([ST])[^P]..cAMP-dependent protein kinase A, basophilic kinaseMOD_ProDKin_1...([ST])P..MAP Kinase, proline directed kinase
* Proteins involved in lipid metabolism, inflammation, redox reactions, lipid transfer, or in membrane complexes.


Just after the III alpha-helix, in group I neurotoxins, a string is present, rich in N and Y, which, in many cases, meets the criteria for recognition by the SH2 domain of NCK1/2, and by phospho-tyrosine binding domains (PTBs). These modules are present in adaptor proteins organizing signaling complexes, by phase separation, on the cytosolic site of the PM [37,38].

Mammalian PLA2G2A possesses two [ST]P motifs, one just before the second α-helix, the second at the C-terminus. In the three-dimensional structure, these two motifs are close to each other (Figure 4B). In myotoxins, and in about half of the group II neuro-myotoxins, the proline of the first motif is conserved but it is preceded by [KR] instead of [ST], and at the C-terminus, although one or more residues of proline are present in many toxins, they are not preceded by [ST] (Table 2). The [ST]–P peptide bond is subjected to isomerization by prolyl isomerase 1 (Pin1) only when the S (or T) is phosphorylated. The S (or T) can be phosphorylated by MAPK kinases, that recognize S/T before a P, or by a cyclin-dependent kinase (CDK), when the P is followed by a basic amino acid (K or R), as in the case of the first [ST]P motif of PLA2G2A. Pin1, by changing the cis/trans conformation of the proline, modulates the behavior of many metabolic factors [39]. Confirming the importance of the proline that precedes the II α-helix, it was already demonstrated that its substitution with alanine profoundly reduced the myotoxic activity of a K49 myotoxin from *Bothrops jararacussu* [40].
Figure 4Main SLiMs that differentiates snake venom PLA2s from their mammalian homologues. (**A**) 3D structures of the snake venom neuro-myotoxin notexin, of human PLA2G1B and their superimposition. The S/T residues (represented in yellow) in the central region of PLA2G1B, belong to four superimposed motifs phosphorylable by GSK3. The tyrosine is represented in magenta form, together with two residues of the pancreatic loop. The motif of interaction with I-BARREL proteins: the loop colored in orange in the group I toxin contains, in numerous cases, an SH2 and PTB binding sites. (**B**) 3D structures of the snake venom myotoxin bothropstoxin-I, of human PLA2G2A and their superimposition. The S/T residue evidenced in yellow in PLA2G2A, when phosphorylated, allows isomerization of the adjacent proline by Pin1. The loop following the second α-helix in the toxin structure, colored in orange, contains a PKA phosphorylation site and an SH2 binding site. The third and last C-terminal amino acid of the toxins (colored in red) forms a PDZ binding motif. The PDB files used for this figure are the same as those described for Figure 1.
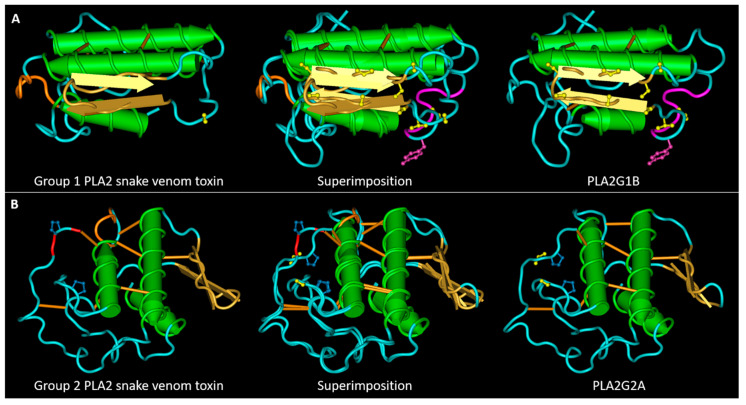



In group II PLA2 toxins, the II α-helix is shorter than that of PLA2G2A and it is followed by an amino acid stretch that, mostly in myotoxins not-D49, is recognized by the cAMP-dependent protein kinase-1 (PKA), and by the SH2 domain of the CRK adapter protein family (Figure 4). PKA has several functions in the cell, including regulating lipid metabolism [41]. The region of PLA2G2A containing the beta-sheet is rich in S/T, although less than in PLA2G1B, and includes a motif that, when the threonine is phosphorylated, binds the forkhead-associated domain (FHA). The FHA domain was discovered in transcription factors, but it is also present in proteins involved in inflammation, such as tumor necrosis factor receptor-associated factor (TRAF), interacting proteins (TIFA and TIFAB) [42], and in the angiogenic factor with G-patch and FHA domains 1 (AGGF1) [43]. Finally, many group II toxins possess the C-terminal pattern [ED]xC that can be recognized by PDZ domains of class 3, other motifs involved in the phase separation of signaling complexes at the PM [38].

## 3. Discussion

The differences in the composition and amino acid sequences of mammalian and snake venom sPLA2s are important to understand to predict the molecular interactions that determine their biochemical actions: the interface with the PM, the formation and breakdown of molecular complexes, and the onset of a protein degradation process that eliminates the sPLA2s themselves.

The interaction with the PM is influenced by the amino acids on the side of the protein containing the active site pocket: positively charged amino acids will attract negatively charged lipid heads and vice versa, while tryptophan residues cause the protein to insert more deeply into the lipid bilayer. PLA2G1B has a tryptophan in the third position that causes the first -helix to penetrate the lipid bilayer, while the other side of the protein, rich in charged amino acids, remains raised, forming an angle with the PM (Appendix A). Group I toxins, on the other hand, have a tryptophan in position 18–19, and one in the C-terminal region (or a Phe), and thus they parallel the PM (Appendix A). Experiments of chemical modifications, conducted on cobra venom sPLA2, have demonstrated that these tryptophan residues are determinant for the interaction with phosphatidylcholine vesicles [44]. However, they may also be important in the interaction with other phospholipids, and with membrane proteins; therefore, they warrant further investigations. Amino acids present in the calcium binding loop and in the active site influence the type of lipid head best suited to fit into the protein pocket [8]. This also applies to catalytically inactive toxins that are nevertheless suitable for retaining lipids in their pocket, as evidenced by many co-crystals of PLA2 such as toxins and fatty acids [45,46] and their affinities for different types of phospholipids. The myotoxin from *B. asper* K49 has a higher affinity for phosphatidic acid [17], a phospholipid involved in the regulation of many cellular signaling pathways [47]. It would be interesting to prove, by site-directed mutagenesis, if the modulation of amino acid in position 49 can influence the specificity of interaction of PLA2s such as proteins with specific phospholipids, and if this interaction is determinant for their toxicity.

Several sPLA2s have been reported to form functional oligomers in contact with the PM, which are useful for the positive regulation of enzyme activity [48,49,50] or protein internalization [26]. There are currently no structural details on these oligomers, but we can assume that they involve interaction between at least two sides of the PLA2, perpendicular to the PM. The most suitable sides appear to be those formed by the C-terminal region, together with the region between the first helix and the calcium-binding loop, and that formed by the central region, the beta sheet, and the loops around it. These regions, in some sPLA2s categories, are enriched in amino acids such as N, Y, S (Figure 3), which are suitable for forming prion-like protein–protein interactions [26,51], while the presence of charged amino acids in these tracts can reinforce the interaction by ionic bonds. The difference in the composition of these regions between mammalian sPLA2s and toxins may therefore indicate a different propensity to form stable oligomers. This could be inquired by performing the site-directed mutagenesis of key charged amino acids, or the swapping of key regions between toxic and non-toxic PLA2s and observing if the mutated proteins conserve their propensity to form oligomers on artificial bilayers or on the cell surface.

sPLA2s oligomers participate in functional molecular condensates on the PM with other peptides and proteins, such as melittin, HSP70, and nucleolin (NCL), which can modulate the enzymatic activity of sPLA2s or even regulate the activity of other enzymes [26,48,50]. For example, sPLA2s can trigger the activity of cytosolic phospholipases and of enzymes that produce oxygenated fatty acid derivatives [52,53,54]. A direct physical interaction among sPLA2s and other PM proteins or enzymes could occur when sPLA2s are internalized. For example, PLA2G2A and cyclooxygenase both localize in the perinuclear region [29]. The parts of sPLA2s containing low complexity/prion-like domains can interact with other proteins containing domains such as NCL and HSP70, while the SLiMs we found in sPLA2s are possible points of interaction with other proteins that can be modulated by post-translational modifications (Table 3). SH2, PTB, and PDZ domains are typical components of adaptor proteins that act as links in protein condensates at the plasma membrane, also involving membrane receptors and integrins [38,55]. The fact that mammalian and toxic PLA2s possess different ligand motives suggests that these proteins participate in different membrane and signaling protein complexes, and snake venom PLA2s can activate the toxic activity of such complexes. The cytotoxicity and the pharmacological activity of the C-terminal peptide of myotoxic group-II snake venom PLA2s [56,57] could also be due to the PDZ binding domain formed by its last three amino acids, and to verify that it would be sufficient to mutate the third-last amino acid.

A particular interesting SLiM is the I-BAR binding motif conserved in the loop that characterizes pancreatic sPLA2s. Although this loop is highly conserved in PLA2G1B, its function has not yet been identified. Mutagenesis of this loop demonstrated that it is not necessary for PLA2G1B catalytic activity [58], but other activities have not yet been investigated. The presence of the I-BAR SLiMs in the loop may suggest that PLA2G1B can participate in membrane protein complexes involved in membrane curvature and cytoskeleton modulation [59].

Finally, the analysis performed in this work provides us with information on the possible mechanisms of cellular control of sPLA2 activity, and of the complexes in which sPLA2s participate. Post-translational modifications can interrupt a process that can become harmful to the cell, and removal of the switch can make sPLA2 uncontrollable and therefore toxic. PLA2G1B has several motifs, not conserved in group I PLA2 toxins, that can be modified by GSK3 which generally has an inhibitory function. GSK3 is known to regulate several enzymes of lipid metabolism; therefore, we think its interaction with phospholipases has a good chance of being real and should be inquired [60,61].

Structural modifications by prolyl isomerase Pin1 can remove sPLA2 from the complexes in which it participates and redirect it towards the degradation system, as happens for many proteins [62,63]. There are no experimental data on the interaction between Pin1 and sPLA2s, but interestingly, both a mouse deprived of Pin1 [39] and a PLA2G2A+ mouse [64] showed resistance to a lipid-enriched diet. The toxicity of myotoxins may be (partly) due to the lack of control by Pin1, and this may explain why the performed mutagenesis of proline in bothropstoxin-I greatly reduces its myotoxic activity [40].

## 4. Conclusions

This comparison of snake venom and mammalian sPLA2 sequence alignments, which, to the best of our knowledge has no precedent in the literature, has revealed several interesting features of these proteins that deserve experimental investigations. The main findings of this analysis are that sPLA2s can be modified, and in this way probably controlled, by kinases (GSK3, PKA, CDK and MAPK), and by Pin1, an [ST] phosphorylation-regulated prolyl isomerase, and that some categories of toxic PLA2s lack the SLiMs involved in these interactions. Moreover, group I snake neurotoxins possess motifs of interactions with PTB, and group II snake myotoxins with SH2 and PDZ domains that are not present in the corresponding homologous mammalian sPLA2s. These interaction domains and post-translational modification motifs can have a role in the formation and breakdown of molecular condensates on the PM, having toxic activities. Further studies will be necessary to unravel the activities triggered by sPLA2s and their molecular partners. Indeed, the in silico analysis proposed in this study will help in the design of further experimental research, suggesting which amino acid trait may be determinant for the activity of sPLA2s, and with which proteins they can interact. The study of SLiMs in sPLA2s opens up the possibility of developing new drugs, both to treat snakebite poisoning and to intervene in the inflammatory and metabolic processes triggered by mammalian sPLA2. Moreover, it extends the prospects for transforming snake venom PLA2s in pharmacological and biotechnological tools. We plan to repeat this study every few years to add to the analysis other phospholipase sequences that will be included in Swiss-Prot, and to look for the presence of other amino acid motifs, not only linear but also three-dimensional, that we have overlooked in this study or that will be discovered in future.

## 5. Materials and Methods

### 5.1. Sequence Collection and Alignment

The sequences of the mono or homomeric snake venom phospholipases A2 were collected from the database UniProt using the string (family: “phospholipase a2 family group I subfamily” keyword: “Toxin [KW-0800]” keyword: “Myotoxin [KW-0959]” AND keyword: “Neurotoxin [KW-0528]” taxonomy: “Serpentes (snakes) [8570]” not (annotation:(type: subunit heterodimer) OR annotation:(type: subunit heterotrimer) OR annotation:(type: subunit heterohexamer))), substituting ‘Group I’ with ‘group II’ to search phospholipases A2 of the second group, and leaving or removing the keywords indicating the type of toxins (myo- or neuro-) to collect the proteins based on their site of action. Only curated entries were considered. To distinguish the catalytically active subfamily, the string (family: “D49 sub-subfamily”) was included with the conjunctions AND or NOT. The sequences of mammalian phospholipases were collected in the database UniProt by entering the gene names PLA2G1B and PLA2G2A, considering only the ten curated entries for PLA2G1B, and the five curated entries plus five other entries in the case of PLA2G2A. The sequence alignment was performed with Clustal Omega (Appendix A) and visualized with the SnapGene viewer software (version 4.3.11) (from Insightful Science, San Diego, CA, USA). Pre- and pro-peptides, when present, were removed from the sequences.

### 5.2. Amino-Acidic Composition Analysis of the β-Sheet Containing Region and C-Terminal Stretch

The amino-acidic composition analysis of the central region (located between the second and third α-helices and including the two β-sheet secondary structures) and of the C-terminal stretch (next to the third α-helix) of PLA2 proteins was performed using the PROTParam tool of Expasy (https://web.expasy.org/protparam/, accessed on 1 March 2021). Amino acids were grouped considering their chemical and biochemical properties as follows: S,T; F,W; D,E; K,R; N,Q; Y and G,A,V,M,L,I,C,H (called “others”). Thus, the frequency of each amino acidic category in the sequence analyzed was obtained by dividing its abundance to the total length of the analyzed sequence and expressing it as a percentage value. Finally, the mean value of the frequency of each amino acidic category was calculated for each family of PLA2A toxins and their mammalian counterparts.

### 5.3. Short Linear Motifs Identification

Short linear motifs (SLiMs) were identified using the ‘The Eukaryotic Linear Motif resource for Functional Sites in Proteins’ (http://elm.eu.org/, accessed on 1 March 2021) [24], considering all cell compartments, choosing ‘Homo sapiens’ as the taxonomic contest with a motif probability cut-off of 100. Only motifs conserved in toxins and not in mammalian phospholipases A2 or vice versa were considered. The syntax used to express amino acid motifs was that adopted by the ELM database and described on the page http://elm.eu.org/infos/help.html (accessed on 1 March 2021).

### 5.4. Tri-Dimensional Structure Representations

The tri-dimensional superimposition of the protein structures was obtained with the ‘Vector Alignment Search Tool Plus’ [65]; pictures and animations were obtained with the ‘Cn3D 4.3.1 3-D structure viewer’.

## Figures and Tables

**Figure 1 toxins-13-00290-f001:**
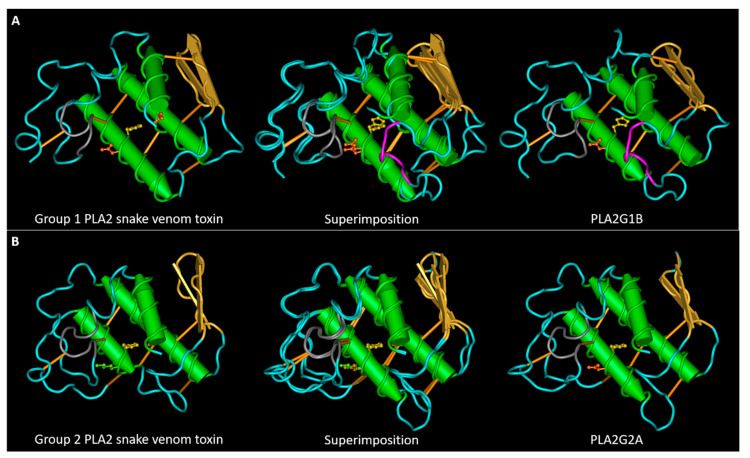
Conservation of the structures of snake venom and mammalian sPLA2s. (**A**) 3D structures of a group I neuro-myotoxin (notexin of *Notechis scutatus scutatus*, PDB entry AE7) and of human PLA2G1B (PDB entry 3ELO) and their superposition. (**B**) 3D structures of a group II myotoxin (bothropstoxin-I of *Bothrops jararacussu*, PDB entry 3I3I) and of human PLA2G2A (PDB entry 1BBC) and their superposition. The calcium binding loop region is colored in grey. PLA2G1B has an extra loop, named the pancreatic loop, represented in magenta. The lateral chains of the amino acids involved in the active site are represented in ball and sticks: H48 in yellow, D49 in orange. Bothropstoxin-I is a PLA2-like toxin (not-D49) with a lysine (in green) in the place of the glutamic acid 49.

**Figure 2 toxins-13-00290-f002:**
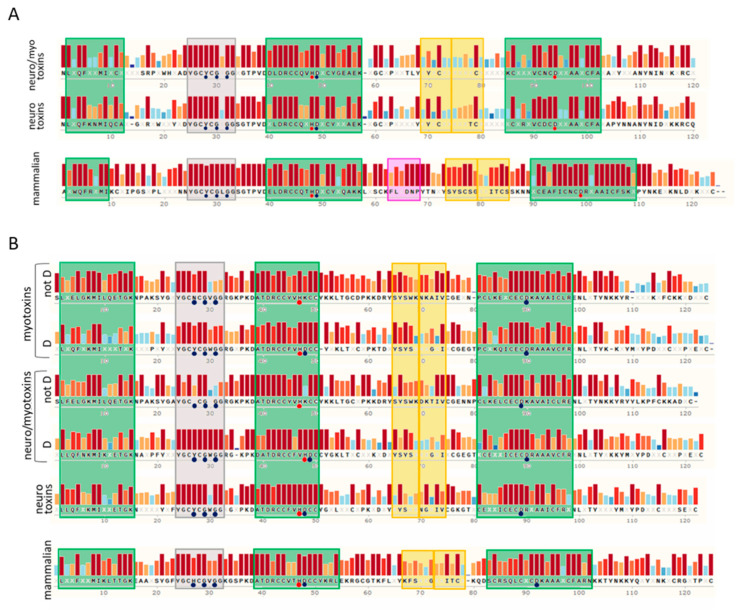
Consensus sequences analyses of snake venom PLA2s and their mammalian homologs. (**A**) Consensus sequences of group I snake venom neuro-myotoxic and neurotoxic PLA2s compared with that of mammalian PLA2G1B. (**B**) Consensus sequences of group II snake venom myotoxic, neuro-myotoxic, and neurotoxic PLA2s compared with that of mammalian PLA2G2A. In the scheme, myotoxins and neuro-myotoxins are divided in the “not D” group and “D” group, according to the absence or presence of the D amino acid in the position known as “49” of the active site, respectively. The sequences were collected from the Swiss-Prot database and aligned with the align tool of UniProt (Clustal O, https://www.uniprot.org/align/, accessed on 1 March 2021). The alignments were then visualized with SnapGene Viewer version 4.3.11. Conserved residues are reported in the consensus sequence, whereas the X code indicates an amino acid. The colored bars above the protein sequence show the degree of conservation, where a red bar indicates the maximum level of conservation (100%). Green, yellow, and grey squares represent α-helix, β-sheet, and calcium binding loop, respectively. The pancreatic loop (pink square, panel A) is present only in mammalian PLA2G1B. Blue circles indicate amino acids involved in the Ca^2+^ binding, whereas red circles represent the catalytic sites. The alignments are reported in the Appendix A.

**Figure 3 toxins-13-00290-f003:**
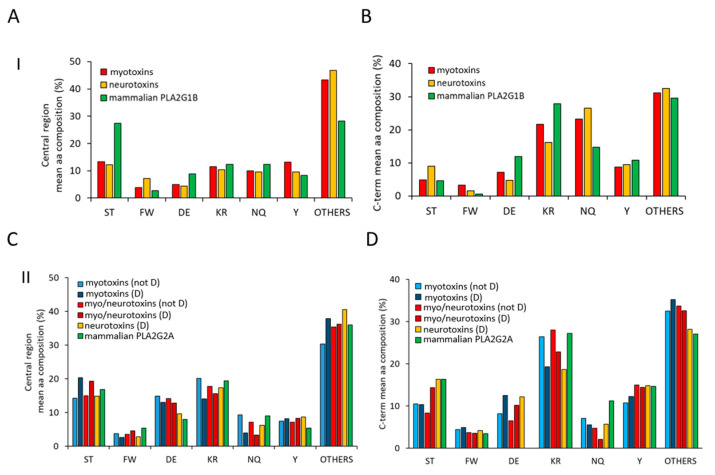
Mean amino acidic composition of the two low complexity regions of sPLA2s. Analyses of the mean amino acidic compositions of the central regions (55–85) (panel **A** and **C**) and of the C-terminal stretches (from 103) (panel **B** and **D**) belonging to toxins of group I and mammalian PLA2G1B (panel **A** and **B**) and to toxins of group II and mammalian PLA2G2B (panel **C** and **D**), respectively. Evaluation of amino acidic composition was performed for each sequence using the ProtParam tool (https://web.expasy.org/protparam/, accessed on 1 March 2021), and then the mean value for each amino acid was calculated and reported in the bar diagram. Amino acids reported in the *x*-axis were separated based on their chemical properties (S,T = phosphorylable sites of ser/thr kinases; FW: aromatic; DE: negatively charged at neutral pH; KR: positively charged at neutral pH; NQ: amine-added amino acids; Y: aromatic phosphorylable sites of Tyr kinases; Others: A, G, V, L, I, H, P, C, M).

## Data Availability

Not applicable.

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
