# Peer review of "Short Linear Motifs Characterizing Snake Venom and Mammalian Phospholipases A2"

_toxins, 2021, doi:10.3390/toxins13040290_

Round 1

Reviewer 1 Report

Comparison of sequence alignments of snake venom and mammalian group I and II phospholipases A2 reveals potential sites of control of their interactions and function

In the present study, the authors collected from UniProt the sequences of snake venom myotoxic and neurotoxic PLA2s, and they compared the alignments of these with those of mammalian group I and II PLA2s, to look for amino acid motifs unique for the different categories. In my opinion, the study is interesting and innovative, including was well delineated. However, I have some comments:

Comment (1): Title is too long so, I recommend the authors to change the title to be more attractive and consistent with the manuscript content.

Comment (2): Unfortunately, the abstract is not clear. The authors should write clear statements about the background, objective and findings in the abstract.

Comment (3): Introduction. There is a brief review of existing knowledge and relevance of study.

Line 50: Add this reference “Snake Venoms in Drug Discovery: Valuable Therapeutic Tools for Life Saving”.

Comment (4): Results. Very good written results. A lot of information is included.

Comment (5): Discussion. Great information in this section but need a proofreading to improve it well.

Comment (6): Materials and Methods. It is clear and the details are sufficient to understand the strategies adopted for the development of the study.

Comment (7): Figures and legends. Legends are adequate and figures are necessary to understand the results obtained.

Author Response

  1. Title is too long so, I recommend the authors to change the title to be more attractive and consistent with the manuscript content.

We thank the reviewer for this suggestion. Accordingly, we have modified the title in ‘Short linear motifs characterizing snake venom and mammalian phospholipases A2’.

  1. Unfortunately, the abstract is not clear. The authors should write clear statements about the background, objective and findings in the abstract.

Re-reading the abstract after some time, we agree with the reviewer's observation. We have therefore rewritten the abstract following the given indications.

  1. There is a brief review of existing knowledge and relevance of study. Line 50: Add this reference “Snake Venoms in Drug Discovery: Valuable Therapeutic Tools for Life Saving”.

Thanking the reviewer for its comment, we add the suggested reference in the introduction section.  

  1. Very good written results. A lot of information is included.

We thank the reviewer for the positive comments on our work, which are very much motivating.

  1. Great information in this section but need a proofreading to improve it well.

Thanks, we have corrected many misspellings.

  1. Materials and Methods. It is clear and the details are sufficient to understand the strategies adopted for the development of the study.

Thanks for this observation.

  1. Figures and legends. Legends are adequate and figures are necessary to understand the results obtained.

Thanks

The parts of the text we have changed in the manuscript are highlighted in blue. 

Reviewer 2 Report

The topic of manuscript is very interesting and the overall organization of it is fine. I have some recommendations, listed as following.

  • The K-49 PLA2 with myotoxicity compared to D49.
  • The C-terminal of K-49 PLA2 is important for some pharmacological/toxic activities, as myotoxicity and inhibition of platelet aggregation. Please, discuss it.
  • What is the importance of such study to develop biotechnological products or drugs. Please, discuss it.
  • Modification of some key aminoacid residues is an interesting way to strengh the relationship structure-function studies, as tryptophan, histidine as well as others. Please discuss it.
  • Supplementary file, please, corret miotoxins.

Author Response

  1. The K-49 PLA2 with myotoxicity compared to D49.

Agreeing with the reviewer, we added a short sentence on the K-49 myotoxicity, page 3, lines 81-82

  1. The C-terminal of K-49 PLA2 is important for some pharmacological/toxic activities, as myotoxicity and inhibition of platelet aggregation. Please, discuss it.

This is another important point. We now discuss about it in “Discussion” section to address the Reviewer’s concerns, page 10, lines 332-335.

  1. What is the importance of such study to develop biotechnological products or drugs. Please, discuss it

Thank you for this reminder. We have now added a short comment about it in “Conclusions” section, page 10, lines 368-371.

  1. Modification of some key aminoacid residues is an interesting way to strengh the relationship structure-function studies, as tryptophan, histidine as well as others. Please discuss it.

As suggested by the reviewer, we now added in each paragraph of the discussion section (if not already reported) some considerations about experimental plans that may be performed using our bioinformatics analysis.

  1. Supplementary file, please, corret miotoxins.

Thank you, we modified the supplementary file accordingly.

The parts of the text that we have changed in the manuscript are highlighted in blue.

Reviewer 3 Report

PLA2, which constitutes a vast number of isozymes with diverse activities, has been the subject of many studies from the viewpoint of structure-function relationships, but these studies have not yet resulted in clear-cut results.

In this background, the discussions in this paper are generally acceptable, but experimental studies to support these discussions are eagerly awaited. In particular, the part that links physiological activity to a single amino acid residue can be quickly verified by mutagenesis experiments.

It would be more convincing if the interatomic distances between the side chain groups of amino acids, which are predicted to interact with the substrate, and the chemical groups of the substrate could be verified not only with a three-dimensional ribbon model but also with an atomic model.

One point: isn't the statement in 175-176 contradictory? Doesn't "low complexity" or “little diversity in their amino acid composition” mean "high conservation among families"?

For example, there are several spelling errors such as "disulphide" in line 51, "venome" in line 52, "catalitic" in line 75, "tird" in line 162, and "signalling" in line 251, 289. A detailed inspection is recommended.

Author Response

  1. In this background, the discussions in this paper are generally acceptable, but experimental studies to support these discussions are eagerly awaited. In particular, the part that links physiological activity to a single amino acid residue can be quickly verified by mutagenesis experiments.

We agree with the reviewer that our in silico analysis strongly demand not only site-, but also region- directed mutagenesis to prove the effective involvement of the identified amino acid motifs. However, to verify the role of the SLiMs present in mammalian PLA2s and not in snake venom PLA2s or vice versa, it will be necessary to mutate both PLA2s, deleting the motifs from one PLA2 and inserting it in the other, so a challenging experimental work. Our aim in this paper was to carry out an accurate bioinformatic analysis of the sequence alignments and to defer verification of some of the amino acid motifs we identified to subsequent works. We believe that our analysis, which to our knowledge has never been carried out before, even if only carried out in silico, will be useful to the entire scientific community working on secreted PLA2s. Moreover, the importance of at least one of the amino acid motifs identified in our analysis has been validated by mutagenesis experiments carried out in the past by Chioato et al., ('Mapping of the structural determinants of artificial and biological membrane damaging activities of a Lys49 phospholipase A2 by scanning alanine mutagenesis).We have now discussed about this point in the last paragraph of the discussion section.

  1. It would be more convincing if the interatomic distances between the side chain groups of amino acids, which are predicted to interact with the substrate, and the chemical groups of the substrate could be verified not only with a three-dimensional ribbon model but also with an atomic model.

In our article, we talked about the interaction between K49 myotoxins and phosphatidic acid. As suggested by the reviewer it would be interesting to analyse the distances between the K49 residue lateral chain and the phosphatidic acid head. Unfortunately, no crystal structure has been determined yet with these two molecules.

  1. One point: isn't the statement in 175-176 contradictory? Doesn't "low complexity" or “little diversity in their amino acid composition” mean "high conservation among families"?

Agreeing with the reviewer, these two statements are contradictory because we used a wrong expression. In the new text we substitute “These two regions are characterized by little diversity in their amino acid composition” with the words “These two regions are characterized by the enrichment of some types of amino acids”.   

  1. For example, there are several spelling errors such as "disulphide" in line 51, "venome" in line 52, "catalitic" in line 75, "tird" in line 162, and "signalling" in line 251, 289. A detailed inspection is recommended.

We thank the reviewer for his careful reading of our article. We have corrected the reported spelling mistakes and further checked the text.

The parts of the text that we have changed in the manuscript are highlighted in blue.

Round 2

Reviewer 1 Report

The Authors addressed all my previous comments in this improved version. I recommend to accept this MS in the present form.